# Orbital Myositis after Herpes Zoster Ophthalmicus: A Case Report and a Narrative Review of the Literature

**DOI:** 10.3390/pathogens13100832

**Published:** 2024-09-26

**Authors:** Edoardo Pace, Guido Accardo, Tommaso Lupia, Maria Felice Brizzi, Silvia Corcione, Francesco Giuseppe De Rosa

**Affiliations:** 1Department of Medical Sciences, Internal Medicine, University of Turin, 10126 Turin, Italy; edoardo.pace@unito.it (E.P.); mariafelice.brizzi@unito.it (M.F.B.); 2Department of Medical Sciences, Infectious Diseases, University of Turin, 10126 Turin, Italy; guido.accardo@unito.it (G.A.); silvia.corcione@unito.it (S.C.); francescogiuseppe.derosa@unito.it (F.G.D.R.); 3Unit of Infectious Disease 2U, University Hospital Città della Salute e della Scienza di Torino, 10100 Turin, Italy

**Keywords:** myositis, herpes zoster, shingles, acyclovir

## Abstract

Herpes zoster ophthalmicus results from the reactivation of the latent varicella zoster virus, affecting the first branch of the trigeminal nerve. In 20–70% of cases, Zoster Ophthalmicus can lead to ocular involvement, affecting various orbital structures. Orbital myositis is a rare but severe complication of herpes zoster ophthalmicus. We present a case of a 52-year-old man with no significant medical history who developed zoster-associated right ocular myositis and dacryocystitis. He was treated with intravenous acyclovir and oral steroids. A review of the literature identified 29 patients across 19 studies. The median age was 61 years, with a slight female predominance. In 55% of cases, the patients had no notable medical history. The most common presentation of myositis involved all oculomotor muscles. There were 22 cases who were treated with intravenous antiviral therapy and 19 received steroids. A full resolution of symptoms was achieved in 51.7% of patients. Zoster-related orbital myositis is a rare complication that should be considered even in immunocompetent individuals. It may occur either before or after the appearance of a vesicular rash. Magnetic resonance imaging is the preferred radiological exam for assessing orbital involvement. Intravenous antiviral therapy should be started within 72 h of symptom onset, and its combination with systemic corticosteroids appears to be an effective treatment for zoster-related ocular myositis.

## 1. Introduction

Varicella zoster virus (VZV) is a human alphaherpesvirus belonging to the genus Varicellovirus and is the causative agent of varicella (chickenpox) and zoster (shingles) [1]. Primary VZV infection commonly occurs in childhood and begins with viral replication in epithelial cells of the upper respiratory mucosa, followed by the widely distributed vesicular rash at the end of an incubation period of 10–21 days.

After primary infection, the virus remains dormant in neurosensory ganglia and may reactivate years or decades later, resulting in vesicular rash in the dermatome that is innervated by the affected ganglion, also known as herpes zoster (HZ) [2]. Reactivation can occur in any situation where the host’s immune system fails to suppress the virus, such as during stress, immunosuppression, or direct trauma [2]. The cutaneous manifestation is often preceded by localized prodromal pain [3], sometimes accompanied by headache, malaise, and disrupted sleep [4]. Reactivation causes a T-cell proliferation along with interferon-alfa and herpes virus-specific antibody production [5]. The most frequent and debilitating complication of HZ is postherpetic neuralgia (PHN), a neuropathic pain syndrome that persists or develops after the dermatomal rash has healed [6].

Herpes zoster ophthalmicus (HZO) occurs when the reactivation of the latent VZV in the trigeminal ganglia involves the ophthalmic division (first branch) of the fifth cranial nerve. HZO accounts for 10% to 20% of herpes zoster cases [2,3,7]. The ophthalmic division of the trigeminal nerve has three major terminal branches (lacrimal, frontal, and nasociliary) which provide sensory innervation to the eyelid, brow, forehead skin, and the skin of the tip of the nose. The nasociliary branch innervates the skin of the tip of the nose and ocular globe, including the cornea and uvea, via its terminal branches called long ciliary nerves. For this reason, Hutchinson’s sign, that is, the presence of a rash at the tip of the nose, is highly correlated with ophthalmic involvement [8].

Approximately 20% to 70% of patients with HZO experience ocular involvement, which may include upper eyelid edema, blepharoptosis, keratoconjunctivitis, anterior uveitis, episcleritis, and scleritis [7,9]. VZV may cause reduced corneal sensitivity, leading to neutrophic ulceration and visual impairment due to scarring.

In very rare cases, HZO may lead to ophtalmoplegia, a rare and self-limiting symptom usually seen in 7–31% of cases. Ophthalmoplegia typically develops within 1–3 weeks of the onset of rash [10] and may be caused by orbital inflammation or neurological complications, presenting as optic neuritis or cranial nerve paresis [11,12]. The oculomotor nerve most commonly involved is the third nerve, while the fourth cranial nerve is least commonly involved [13]. Orbital myositis may also cause a restriction of ocular movement, leading to ophthalmoplegia [14]. The pathogenesis of nerve or muscle inflammation can be attributed either to the direct cytopathic effect of the virus or to a reactive immunological response.

The aim of this study is to report our experience with a case of orbital myositis by VZV and to narratively review the literature on this rare but serious manifestation of HZO.

## 2. Materials and Methods

The current narrative review followed the Scale for the Assessment of Narrative Review Articles (SANRA) checklist and we have included a flow-chart of the studies revised (Figure 1) [15].

A search was run on PubMed using the terms (‘Myositis’ [Mesh]) AND (‘Herpes’ [Mesh]) and (‘Myositis’ [Mesh]) AND (‘Zoster’ [Mesh]) in English. Studies were filtered for practice guidelines, guidelines, meta-analyses, systematic reviews, narrative reviews, case series, and case reports. Therefore, we have filtered results including only humans and adult patients.

Our search strategy permitted the identification of 110 papers, of which 88 were excluded by title and abstract evaluation. The timeframe of the included studies were 1 January 1980 to 1 July 2024. Then, the reviewers studied titles and abstracts. Subsequently, 24 papers were included. Finally, a quality assessment of full-text studies was performed by two independent reviewers (A.G. and E.P.). Researchers reviewed the summary of all articles sought and ultimately used data from full articles to compile this review paper. Researchers assessed the inclusion of all titles and abstracts without language limitations in English. The criteria for excluding papers were as follows: written in a language other than English, duplicates of previously included studies, papers with no methods described in the text, papers not strictly related to the aim of the study.

The quality of the studies included were assessed according to the SANRA checklist [15] and the six items which form the revised scale are as follows: an explanation of the importance and the aims of the review; a search of the literature and referencing; and a presentation of the evidence level and relevant endpoint data and results of the process, as reported in Figure 1.

We performed descriptive statistics on the entire study population. Data were analyzed using standard statistical methods. Variables were described with medians, absolute values, and rates.

## 3. Case Report

A 52-year-old man presented to the emergency department within two days of increased right-eye lacrimation, photophobia, headache, and right forehead and eyebrow vesicles. His medical history was unremarkable, and he was not on any chronic medication. The patient was afebrile and did not exhibit diplopia or visual impairment. The right-eye ophthalmic examination revealed ptosis, conjunctival hyperaemia, hyporeactive mydriasis, and increased intraocular pressure. No signs of uveitis, corneal erosions, scleral involvement, or retinal involvement were detected. Left-eye examination was normal. Cranial CT and chest x-ray were negative. Routine biochemical and hematological examinations were within normal limits. An ocular conjunctiva swab with qualitative polymerase chain reaction (PCR) turned positive for VZV-DNA. He was treated with intravenous Acyclovir (10 mg/kg, three times daily) for two days and was discharged with oral Valacyclovir (1 g, three times daily) for ten days.

Two weeks later, the patient returned to the emergency department due to worsening symptoms and the onset of diplopia. The clinical presentation with skin vesicles, erythema, facial edema, right-eyelid and eye lacrimation is shown in Figure 2.

Ophthalmic examination revealed increased eyelid edema, ptosis, and conjunctival hyperaemia, with no involvement of uveal or corneal structures. Orthoptic assessment of the right eye detected limited elevation and left lateral vision, resulting in both vertical and horizontal diplopia (Figure 3). Cranial and pituitary CT scan did not show signs of inflammation of the oculomotor muscles. Magnetic resonance imaging (MRI) of the right orbit (Figure 4) revealed myositis affecting the superior rectus, medial rectus, lateral rectus, and superior oblique muscles, along with lacrimal gland enlargement and minimal thinning of the optic nerve. No abnormalities in the arterial and venous vascular structures were identified. Routine blood tests remained normal, with a slight increase in VZV IgM and IgG. A second cutaneous swab on the vesicles for VZV was not performed due to the clinical presentation and the recent PCR performed in the recent hospitalization. HIV antibody and antigen tests were negative and lymphocyte subpopulations were within normal ranges.

Intravenous Acyclovir (10 mg/kg, three times daily) was administered for a total of two weeks. After seven days of antiviral therapy, the patient showed limited improvement, prompting the initiation of oral steroid therapy (Prednisone 25 mg daily), which resulted in subsequent symptom relief. Oral Pregabalin (75 mg twice daily) was also started. He was discharged after two weeks with a regimen of oral antiviral therapy (Valacyclovir 1 g, three times daily, with a tapering dose over two weeks) and a gradual reduction of oral steroids.

One month after discharge, MRI showed complete resolution of the right-eye myositis, improvement of dacryocystitis, and no alterations of the optic nerve (Figure 4). From the clinical point of view, ocular movements had returned to normal, as illustrated in Figure 3. On ophthalmic examination, minimal residual right ptosis and complete resolution of eyelid edema, lacrimation, and conjunctival hyperemia were observed.

At the three-month follow-up, ophthalmic evaluation revealed no diplopia or ptosis, with the complete resolution of orbital inflammation. However, the patient reported persistent right-frontal paraesthesia, having discontinued Pregabalin due to low tolerance.

## 4. Results

The search strategy identified 110 articles, of which 19 studies were included for the final analysis, with a total of 29 patients, and data were showed in Table 1. 

In the population of patients collected from the study, the median age was 61 (13–81) years, with a slight prevalence of the female sex (N = 17; 58.6%). Six studies did not report any information regarding comorbidities. Among studies that reported medical history, 12 of 22 patients had no comorbidities (55%). The most frequent comorbidity was diabetes mellitus (N = 5). Moreover, Temnogorod et al. [16] reported comorbidities in only two of seven patients: one with chronic lymphocytic leukemia and one with HIV. Patients with orbital myositis complicating HZO also had skin (N = 18; 81.9%), neurological (N = 17; 77.3%), or ocular involvement (N = 13; 59.1%). Cases of orbital myositis were diagnosed using head and orbital MRI (N = 15; 51.7%), CT (N = 10; 34.5%), or both imaging examinations (N = 4, 13.8%). The anatomical distribution of myositis among orbital muscles is very heterogeneous. In most cases, all orbital muscles (N = 8) were involved in the inflammatory process. Secondly, medial (N = 5) and lateral (N = 5) muscles were frequently involved. Moreover, three patients reported an involvement of the lacrimal gland (i.e., dacryocystitis).

From the therapeutic point of view, patients were treated with oral antiviral alone (N = 5, 17.2%); oral antiviral switched to the intravenous (IV) route (N = 2, 6.9%); or IV from the beginning of the treatment (N = 22, 75.9%). Nineteen patients have received steroids (55.2%), either orally (N = 7) or intravenously (N = 12).

After treatment, full resolution was reached in 51.7% of patients (N = 51).

**Table 1 pathogens-13-00832-t001:** Clinical, radiological, treatment, and follow-up characteristics of the patients collected from the literature.

Author	Age	Sex	Immune Impairment	Clinical Presentation	Imaging	Radiographic Findings	Treatment	Status at Last FU	FU Time
Volpe et al. [14]	45	M	NR	Left eye pain, diplopia, proptosis.	CT	Enlargement of the medial, lateral and inferior rectus muscles in the left orbit.	Acyclovir OS + topical steroids	Full resolution	12 months
Tseng Y. et al. [17]	54	M	Cirrhosis	Left orbital pain; mild proptosis and hyperaemic conjunctiva on the left side.	MRI	Enlargement of the medial and lateral rectus muscles in the left orbit, enhanced with gadolinium.	Acyclovir IV	Full resolution	4 months
Temnogorod et al. (N = 7) [16]	70 (47–84)	F (6); M (1)	Chronic lymphocytic leukeamia (1), HIV (1)	Proptosis, blepharoptosis, ophthalmoplegia, diplopia, and visual loss.	CT (3), CT + MRI (3), MRI (1)	Myositis (no reported specific muscular involvement) (7); optic nerve sheath enhancement (1); encephalitis (1); dacryocystitis (2).	IV acyclovir (5), IV acyclovir + IV Dexamethasone (1), IV Acyclovir + Methylprednisolone (1)	Complete resolution (3), improvement (4)	Median 7 months (3–21)
Pereira et al. [18]	89	F	NR	Vesicular rash in the left V1 distribution, with a positive Hutchinson sign.	CT + MRI	Enlarged lateral, inferior, and medial recti muscles and proptosis OS.	Acyclovir IV + oral Prednisone	Full resolution	11 months
Patheja et al. [19]	60	M	Previous mantle cell lymphoma	Left eye pain, oedematous eyelids, conjunctival injection, proptosis.	MRI	Myositis, high signal intensity of the retrobulbar fat and linear signal change of the optic nerve head and sheath, dacryocystitis.	First, oral Valacyclovir + topical steroids, then, due to the worsening, Acyclovir IV with pulsed intravenous methylprednisone	Visual field remained markedly restricted	6 months
Paraskevas et al. [20]	67	F	NR	Intense pain in the left V1 area, left ptosis of the left upper eyelid, and diplopia.	MRI	Fusiform enlargement and paramagnetic enhancement of the left orbital muscles. The trigeminal nucleus and nerve show increased sign intensity.	Acyclovir IV + methylprednisolone IV	Left eye saccades still appearing slowed	5 months
Merino-Iglesias et al. [21]	61	M	NR	Pain in his left eye, left facial edema associated with vesicles.	CT	Increased thickness of the medial and superior rectus.	Acyclovir IV and oral methylprednisolone	Full resolution	12 months
Loubet et al. [22]	70	M	NR	Right periorbital pain vesicular rash in the area of the right trigeminal nerve (V-1) associated with ptosis and complete homolateral opthalmoplegia.	MRI	Inflammation of the right extraocular muscles, exopthalmia, enhancement of the right trigeminal (V).	Oral corticosteroids + Acyclovir IV	Full resolution	1 month
Lee Cy et al. [12]	78	M	DM	Pain and swelling of the left eyelid; patchy erythema with clustered vesicles.	MRI	Left extraocular muscles (arrows) on T2; mild enhancement of the left optic nerve sheath (arrowhead), left posterior orbital wall, and left orbital apex on contrast-enhanced T1-weighted images.	Acyclovir IV + NSAIDs at first, then Acyclovir IV + Prednisolone	Limitation of abduction and paralysis of the left upper eyelid.	6 months
Krasnianski et al. [23]	67	F	NR	Right-sided edema of the eyelid, scleral hyperaemia, vesicular lesions in the area of the ophthalmic nerve, and erythema of the periorbital skin.	MRI	Oedematous thickening and gadolinium enhancement in all right-sided external ocular muscles and in the right optic nerve. Contrast enhancement of the retrobulbar fat and periorbital soft tissues on the right side. Cerebral MRI, particularly of the nuclear areas of III, IV, and VI cranial nerves and cavernous sinus, were unremarkable.	Acyclovir IV + Prednisolone IV	NR	NR
Kim HT et al. [24]	66	M	DM	At first, left orbital shooting pain, mild ptosis and hyperaemic conjunctiva. After 3 days, he developed vesicular skin rashes at the distribution of the ophthalmic branch of the left trigeminal nerve and along the nasal ridge.	CT	Left orbital myositis involving superior, inferior, medial, and lateral rectus muscles.	Acyclovir IV, IV dexamethasone	NR	5 months
Kawasaki et al. [25]	47	F	NR	Left retro-bulbar pain; in the next 3 days, mild ptosis and vesicular cutaneous rash in the distribution of the ophthalmic division of the left trigeminal nerve.	MRI	Enlargement of all the extraocular muscles on the left side. Contrast enhancement of the extraocular muscles, orbital fat, and periorbital soft tissues.	Acyclovir IV + oral Prednisone	Minimal proptosis	2 months
Daswani et al. [26]	67	F	DM	Ptosis, healed pigmented scars on the right side of forehead and lid. Extraocular movements were restricted on levo- and dextro-elevation.	MRI	Asymmetric thickening of the right extraocular muscles, suggestive of orbital myositis, and thickening of the right oculomotor nerve.	Oral Acyclovir	Improvement of visual acuity, resolving ophthalmoplegia	1 month
Conrady CD et al. [27]	13	F	None	Right upper and lower eyelids, pain with eye movements, rash in the right face.	CT	Right periorbital swelling and asymmetric enlargement of the right inferior and medial rectus muscles. Thickening of medial and inferior rectus muscles.	Acyclovir IV; oral Prednisone.	Full resolution	10 days
Chiang et al. (N = 2) [28]	70 (63–77)	M (1); F (1)	None	Pain OS with mild swelling of her left upper eyelid; significant swelling of the left eyelids, cheek, and temple. Small nonspecific violet-colored skin lesions.	CT (2)	Enlargement of the lateral rectus muscle. Left proptosis and marked preseptal and subcutaneous edema with mild, nonspecific stranding in the left orbit.	Acyclovir IV + oral Prednisone (1); Acyclovir IV (1)	Full resolution (2)	NR
Chen et al. [29]	56	M	None	Right-sided frontal and supraorbital headache, with a dull throbbing; right-sided ptosis.	MRI	Diffuse right-sided orbital myositis.	Oral valacyclovir	Full resolution	NR
Chandrasekharan et al. [30]	60	F	DM	Reduced vision in the left eye with complete left ptosis and limited eye movements.	MRI	Enhancement of the optic nerve and perineural sheath, extraocular muscles and soft tissues.	IV methyl prednisolone + oral Acyclovir	Mild residual ptosis, limitation of eye adduction, elevation and depression	3 months
Bak et al. (N = 4) [31]	57 (32–69)	M (2); F (2)	DM (1)	Zoster rash, pain upon left ocular movement, orbital pain.	MRI (4)	Optic nerve enhancement, rectus muscle enlargement	Acyclovir IV + IV methylprednisolone (3), oral famciclovir, IV dexamethasone (1)	Full resolution (3), residual ptosis, proptosis	6.75 months (2–17)
Badilla J et al. [32]	81	F	NR	Right retrobulbar pain, proptosis, and visual loss.	CT	massive enlargement of the right oblique muscles and moderate enlargement of recti muscles.	At first, oral valacyclovir + oral Prednisone, then, due to worsening conditions, IV acyclovir + oral Prednisone	Mild motility, restriction of right eye movements in all directions of gaze	1 month

Abbreviations: NR: not reported; CT: computed tomography; IV: intravenous; F: female; M: male; DM: diabetes mellitus; MRI: magnetic resonance imaging; FU: follow-up; V1: first branch of the fifth cranial nerve.

## 5. Discussion

The purpose of this study is to present a rare case of ocular myositis following HZO and to review the literature available in this field. 

In our review, the median age of patients experiencing orbital myositis was 61 years with a slight female predominance. In the case here reported, we have observed a middle-aged man, in line with data reported in the literature. Specific comprehensive data on orbital myositis epidemiology are lacking, and most of the information is available from case reports and case-series [12,14,16,17,18,19,20,21,22,23,24,25,26,27,28,29,30,31,32]. Regarding the epidemiology of HZ and HZO, the incidence of both increases with age, likely due to a decline in VZV-specific cellular immunity [33,34]. However, another study indicated that the average age of HZO onset is decreasing (from 61 years in 2007 to 56 years in 2013) [35]. The epidemiology of HZ and HZO could change in the next years due to vaccination strategies; in fact, vaccinating adults is critical to reduce the risk of vaccine-preventable diseases, especially for those with immunocompromising health conditions [36]. HZO is more common in women than men [33,37], with annual incidence estimated at 44.5 versus 33.1 cases per 100,000 person-years [34]. These data suggest that patients with HZ-associated myositis share similar characteristics with the general population affected by HZ and HZO. 

Moreover, uncommon manifestations (i.e., acute retinal necrosis, optic neuritis, orbital apex syndrome, pan-uveitis, and cellulitis complicated by sepsis) are significantly associated with immunosuppression and diabetes [38]. However, it is noteworthy that the majority of recorded patients (55%) had no significant pathological medical history, and only five had diabetes. This indicates that HZO myositis should be suspected and studied even in patients without comorbidities or immunosuppressive conditions. The patient presented in our case report showed an unremarkable medical history, as did most of the patients collected in the narrative review.

The literature reports that the mean time to the onset of ocular involvement is 1.8 weeks (range 1–4 weeks) after the vesicular eruption [25]. However, some cases have documented the onset of painful ophthalmoplegia before the rash, with a mean latency of 4.8 days (range 2–7 days) [18]. In our case, the patient developed forehead and eyebrow vesicles as an early symptom, along with conjunctivitis and periorbital edema, while myositis and the resulting diplopia emerged two weeks after the initial symptoms. It is important to emphasize that orbital inflammation and myositis may be the first signs of HZO, even in the absence of a vesicular rash [31]. Furthermore, HZO was linked to a wide spectrum of clinical presentations and complications, as reported in Figure 5.

In our review, the majority of the patients (65.5%) were evaluated with head and orbital MRI, either alone or in addition to CT imaging. In our case, orbital myositis was detected by MRI but not by CT. It is important to note that the MRI was conducted 7 days after the CT scan; nevertheless, MRI appears to be more sensitive and accurate for orbital assessment. For this reason, we recommend using MRI for evaluating ocular involvement and complications in cases of HZO.

Ophthalmoplegia occurs in approximately 3.5–10.1% of HZO cases [23]. Previously, this manifestation was attributed to nerve involvement, but many recent reports have described HZO-associated oculomotor myositis as a cause of diplopia and ophthalmoplegia [16]. In our study, the involvement of all oculomotor muscles was the most common presentation of myositis and this clinical and radiological finding was also observed in our patient. Additionally, in the case here presented, MRI did not reveal any abnormalities in the oculomotor nerves, though some of the literature reports documented inflammation affecting both muscles and nerves [16,20]. Furthermore, ophthalmoplegia is generally described as a self-limiting complication of HZO [26]. A literature review found that the complete or near-complete resolution of ophthalmoplegia occurred in 65% of HZO cases within a range of 2 weeks to 1.5 years [39]. In our study, we observed a lower percentage of full resolution (51.7% of patients, follow-up range 10 days–21 months). Another case series on HZO-related acute orbital syndrome reported that an improvement in orbital signs can be expected after antiviral therapy, although incomplete resolution is not uncommon [16]. This variability is likely due to differences in the timing of antiviral therapy initiation, which is one of the main predictive factors for complete recovery [33].

Moreover, our patient developed acute dacryocystitis, a rare complication of HZO, with only eight cases reported in the literature [40]. Lacrimal gland involvement may be underreported due to being overshadowed by other HZO manifestations [16]. 

At follow-up, our patient experienced PHN, defined as significant dermatomal pain persisting for 3 months or more after the onset of rash, often accompanied by symptoms such as allodynia, reduced sensation, and paresthesia [33]. PHN is the most common chronic complication of HZ infection, described in 9–45% of cases [2], and is associated with the severity of the rash, the presence of ocular involvement, decline in corneal sensation, and early severe neuralgia [7]. While antiviral therapy reduces the severity and duration of HZ skin involvement, it does not influence the development of PHN [41].

The treatment of HZO aims to reduce the intensity and duration of symptoms, as well as to prevent recurrence and complications, particularly those affecting ocular structures [33]. Ophthalmologic consultation and the early initiation of antiviral therapy (within 72 h of the onset of symptoms) is recommended for all patients with HZO to minimize the risk of complications [42]. 

Oral treatment options include Acyclovir (800 mg, five times a day), Famciclovir (500 mg, three times a day), and Valaciclovir (1 g, three times a day), which have been shown to all be similarly effective [43,44]. However, intravenous (IV) antiviral therapy (Acyclovir, 10 mg/kg three times daily) is recommended for disseminated or complicated HZO and in immunocompromised patients [33]. Previous studies suggest that IV antiviral therapy may be beneficial for patients with HZO-associated ocular involvement [16,45]. In our study, 82.6% of patients received IV acyclovir, with 22 starting on IV treatment and 2 switching from oral therapy. Our patient was initially treated with a brief course of IV therapy, followed by an oral regimen. However, he developed complications and ocular involvement while on oral antiviral therapy. According to the literature, while there is no optimized treatment proven by clinical trials, IV antiviral therapy is recommended for patients with complicated HZO or at risk of developing ocular involvement.

Our research did not reveal a general consensus on the duration of antiviral therapy. For uncomplicated HZO, the standard duration of antiviral therapy is 7–10 days [2]. However, in the case of HZO-related orbitopathy, IV Acyclovir is recommended at a dose of 10–15 mg/kg three times daily for 2–3 weeks [33]. Our patient was treated with IV Acyclovir (10 mg/kg, three times daily) for two weeks, followed by oral Valacyclovir (1 g, three times daily), with a tapering dose over the next two weeks. The use of long-term suppressive therapy with oral antiviral agents remains controversial, as there are no data from randomized, placebo-controlled studies to support this practice [33].

Oral and topical ophthalmic corticosteroids may be prescribed alongside antiviral agents as anti-inflammatory treatments, as they have been shown to reduce the duration of pain during the acute phase of HZO [46]. However, European guidelines on HZO management recommend the use of topical and systemic steroid therapy only in patients who present with acute retinal necrosis [44]. Furthermore, topical corticosteroids should be used judiciously and in direct consultation with an ophthalmologist, as their application can lead to HZO-related complications such as stromal keratitis, uveitis, or scleritis/episcleritis [2]. It is also important to note that corticosteroids should not be used as monotherapy, as their immunosuppressive effects can promote viral replication, potentially resulting in complications like retinal necrosis if not paired with antiviral agents [44].

As for retinal necrosis, systemic steroid therapy appears to be beneficial for HZO-associated myositis due to its anti-inflammatory effect [31]. In our study, 55.2% of patients received systemic corticosteroids, either orally (N = 7) or intravenously (N = 12). In our specific case, initiating oral steroid therapy (Prednisone 25 mg daily) in addition to IV Acyclovir led to symptom relief and an improvement in clinical and radiologic findings. Although the treatment of HZO-related myositis is not well-established due to the rarity of this condition, the combination of intravenous antiviral agents and systemic corticosteroid therapy appears to be the most commonly used and effective treatment option.

The limitations of the study were primarily the narrative methodology of the study; moreover, most of the studies collected were case reports or case-series with a low number of patients included. Secondly, we mentioned the potential for bias in the article selection performed independently by two different authors. Finally, a limitation is the lack of high-quality studies on orbital myositis.

## 6. Conclusions

Orbital myositis is a rare but serious complication of HZO leading to ophthalmoplegia and diplopia. It should be suspected and studied even in patients without comorbidities or immunosuppressive conditions. Typically, this manifestation follows the vesicular rash on the cutaneous area of the ophthalmic branch of the trigeminal nerve, but in some cases orbital inflammation and myositis may occur in absence of a vesicular rash. MRI is recommended for the radiological evaluation of HZO-related ocular involvement and complications. Improvement in orbital signs can be expected after antiviral therapy, although complete resolution is observed in 50–60% of cases. Early antiviral treatment of HZO (within 72 h of the onset of symptoms) can reduce the intensity and duration of symptoms and help prevent recurrence and ocular complications. Although there is no optimized treatment of HZO-related myositis proven by clinical trials, intravenous antiviral agents in conjunction with systemic corticosteroid treatment appears to be effective for this rare manifestation of HZO.

## Figures and Tables

**Figure 1 pathogens-13-00832-f001:**
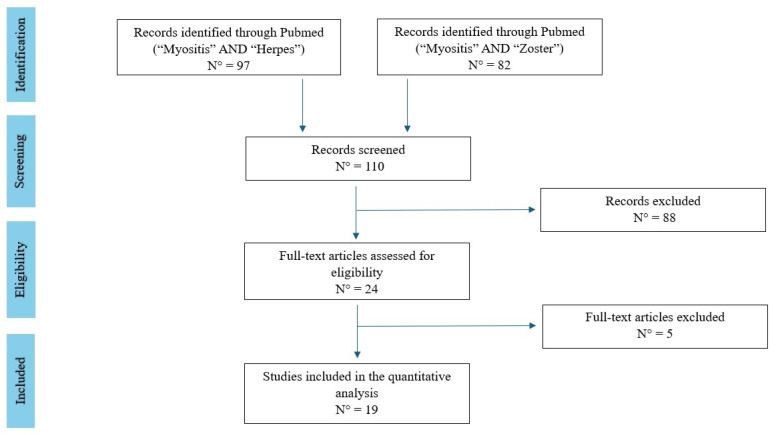
Flow-chart of the studies revised in the narrative review.

**Figure 2 pathogens-13-00832-f002:**
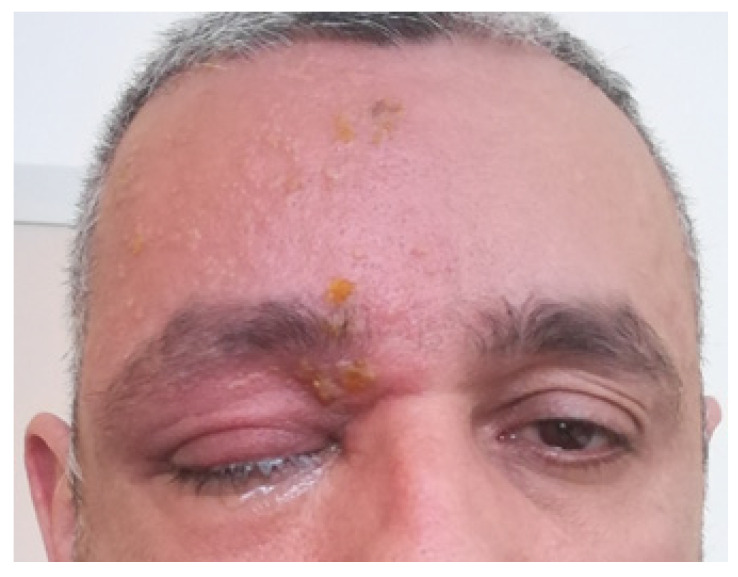
Clinical presentation at admission in hospital ward.

**Figure 3 pathogens-13-00832-f003:**
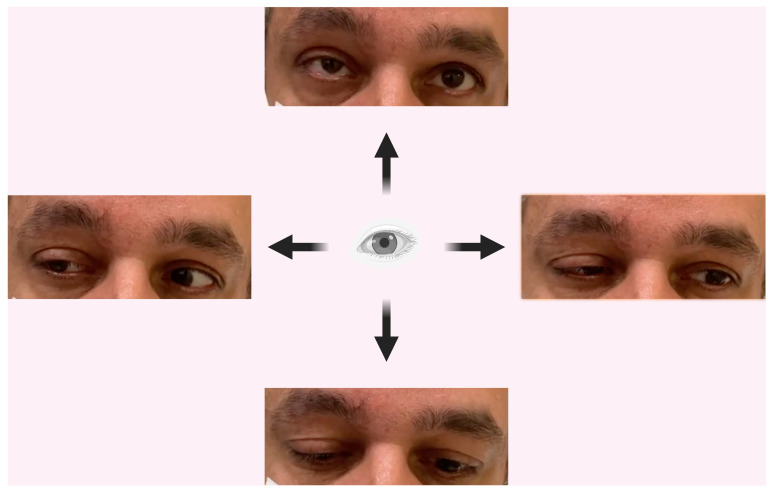
One month of follow-up after hospitalization.

**Figure 4 pathogens-13-00832-f004:**
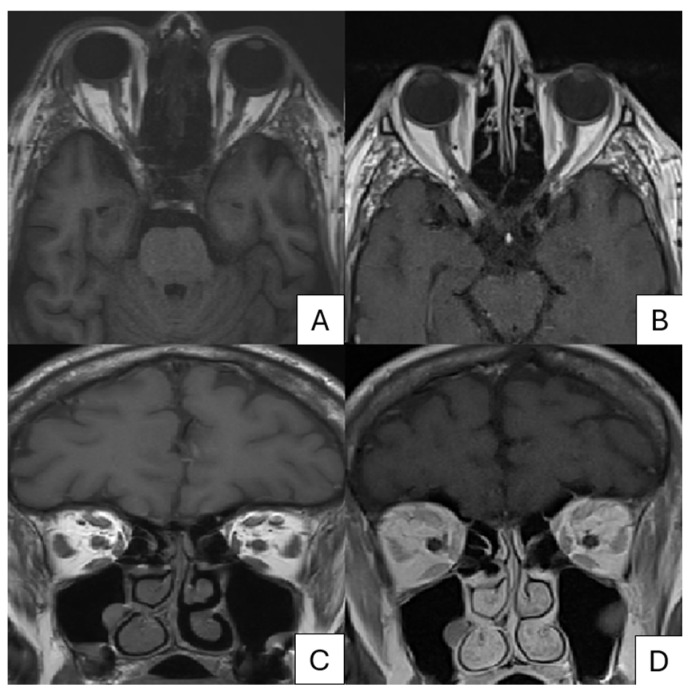
(**A**,**C**) Transverse and Coronal T1-weighted orbit MRI of patient with altered signal intensity of right superior rectus, medial rectus, lateral rectus, and superior oblique muscles. At one-month follow-up (**B**,**D**), improvement of extraocular muscle myositis can be seen on T1-weighted transverse and coronal image of orbits.

**Figure 5 pathogens-13-00832-f005:**
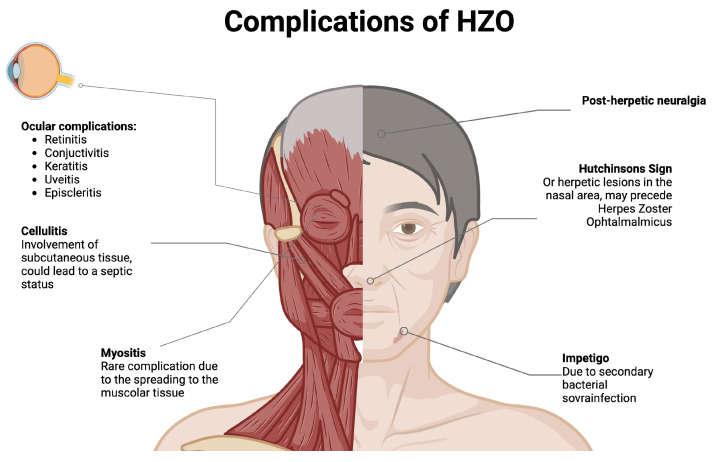
HZO-related cutaneous and ocular presentations and complications.

## Data Availability

Data will be available following reasonable request to the authors.

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
