# Peer review of "Orbital Myositis after Herpes Zoster Ophthalmicus: A Case Report and a Narrative Review of the Literature"

_pathogens, 2024, doi:10.3390/pathogens13100832_

Round 1

Reviewer 1 Report

Comments and Suggestions for Authors

Specific Comments for Revision:

Line 72-73: The manuscript mentions that the review followed the "Scale for the Assessment of Narrative Review Articles (SANRA) flow-chart (Figure 1)." SANRA is a checklist for evaluating narrative reviews, not a flowchart. The review should be revised to reflect that it followed SANRA principles and that the flowchart was an additional organizational tool. This change will accurately represent SANRA’s purpose and clarify the methodology used in the review.

While there are no strict guidelines for conducting narrative reviews, using a tool like SANRA adds transparency to the methodology. To further improve transparency, the authors should provide clear explanations for excluding certain articles. I recommend adding a section that details the exclusion criteria and the reasons for excluding specific studies from the final review.

Materials and Methods: The quality of the studies included in the review has not been formally assessed.

Line 76-80: Please indicate the timeframe of the studies included in the review.

Ethical Considerations: There is no mention of ethical considerations or anonymization in the manuscript. If patient data or case reports are included, an ethical statement should be added, indicating that patient consent was obtained, and all identifying details were anonymized.

Limitations Section: A limitations section is missing. It is important to acknowledge the limitations of this narrative review, such as the potential for bias in article selection, reliance on case reports, and lack of high-quality studies on orbital myositis.

The manuscript covers an interesting and important topic, but it requires a major revision to improve its scientific rigor, transparency, and overall structure.

Author Response

Line 72-73: The manuscript mentions that the review followed the "Scale for the Assessment of Narrative Review Articles (SANRA) flow-chart (Figure 1)." SANRA is a checklist for evaluating narrative reviews, not a flowchart. The review should be revised to reflect that it followed SANRA principles and that the flowchart was an additional organizational tool. This change will accurately represent SANRA’s purpose and clarify the methodology used in the review.

Dear reviewer, thank you for your comments that improve notably our manuscript. We agree with you, despite that is only an error of wording, we have followed SANRA checklist. We have checked the test, and the checklist and changed the text according to your suggestions.

While there are no strict guidelines for conducting narrative reviews, using a tool like SANRA adds transparency to the methodology. To further improve transparency, the authors should provide clear explanations for excluding certain articles. I recommend adding a section that details the exclusion criteria and the reasons for excluding specific studies from the final review.

Dear reviewer, thank you for your comments that significantly improved our manuscript. We have added a section with exclusion criteria in the text according to your suggestions.

Materials and Methods: The quality of the studies included in the review has not been formally assessed.

Dear reviewer, thank you for your comments that significantly improved our manuscript. We have added a section with the quality of the studies included in the review according to your suggestions.

Line 76-80: Please indicate the timeframe of the studies included in the review.

Dear reviewer, thank you for your comments that significantly improved our manuscript. We have added the timeframe of the studies included in the review.

Ethical Considerations: There is no mention of ethical considerations or anonymization in the manuscript. If patient data or case reports are included, an ethical statement should be added, indicating that patient consent was obtained, and all identifying details were anonymized.

Dear reviewer, thank you for your comment. Informed consent was obtained as reported in the section “informed consent section” and uploaded at the submission of the article.

Limitations Section: A limitations section is missing. It is important to acknowledge the limitations of this narrative review, such as the potential for bias in article selection, reliance on case reports, and lack of high-quality studies on orbital myositis.

Dear reviewer, thank you for your comments, which significantly improved our manuscript. According to your suggestions, we have added a section with study limitations in the text.

Reviewer 2 Report

Comments and Suggestions for Authors

The manuscript describes a case report of a patient with HZO and a literature review. The manuscript is organized and well written.

Major point:

While the clinical manifestations are well described there is only a brief mention of the medical history as unremarkable. It is not mentioned whether the patient had a history of previous zoster. Also, the laboratory tests are only briefly mentioned as "routine blood tests remained normal". It is not mentioned which blood test that were taken. Surprisingly, no samples were taken from the vesicles for testing VZV DNA by PCR. Only a serological blood test was done to assess VZV IgG and IgM. This test is of limited use as the seroprevalence to VZV is very high. How could the authors be absolutely sure that the symptoms are caused by VZV and not herpes simplex virus? This needs to be clarified.

Minor points:

Figure 4. The figure legend could be altered in order to clarify. It reads "(A-C).... (B-D) and suggest that it should read "(A and C ... (B and D).

Line 204. The wording "forehand" should be " forehead" presumably.

Author Response

The manuscript describes a case report of a patient with HZO and a literature review. The manuscript is organized and well written.

Major point:

While the clinical manifestations are well described there is only a brief mention of the medical history as unremarkable. It is not mentioned whether the patient had a history of previous zoster. Also, the laboratory tests are only briefly mentioned as "routine blood tests remained normal". It is not mentioned which blood test that were taken. Surprisingly, no samples were taken from the vesicles for testing VZV DNA by PCR. Only a serological blood test was done to assess VZV IgG and IgM. This test is of limited use as the seroprevalence to VZV is very high. How could the authors be absolutely sure that the symptoms are caused by VZV and not herpes simplex virus? This needs to be clarified.

Dear reviewer, thank you for your comments that significantly improved our manuscript. We agree with you. During the second hospitalization, a second swab was not performed. In the first admission, a conjunctiva swab turned positive for VZV-DNA. We have reported this data in the text.

Minor points:

Figure 4. The figure legend could be altered in order to clarify. It reads "(A-C).... (B-D) and suggest that it should read "(A and C ... (B and D).

Dear reviewer, thank you for your comments that improve notably our manuscript. We have changed the text according to your suggestions.

Line 204. The wording "forehand" should be " forehead" presumably.

Dear reviewer, thank you for your comments that improve notably our manuscript. We have changed the text according to your suggestions.

Round 2

Reviewer 1 Report

Comments and Suggestions for Authors

All comments have been adequately addressed.

Reviewer 2 Report

Comments and Suggestions for Authors

I am satisfied with the author´s response to my questions.